# INTEGRATING STATE SPACE MODEL AND TRANSFORMER FOR GLOBAL-LOCAL PROCESSING IN SUPER-RESOLUTION NETWORKS

## ABSTRACT

Single image super-resolution aims to recover high-quality images from low-resolution inputs and is a key topic in computer vision. While Convolutional Neural Networks (CNNs) and Transformer models have shown great success in SISR, they have notable limitations: CNNs struggle with non-local information, and Transformers face quadratic complexity in global attention. To address these issues, Mamba models introduce a State Space Model (SSM) with linear complexity. However, recent research shows that Mamba models underperform in capturing local dependencies in 2D images. In this paper, we propose a novel approach that integrates Mamba SSM blocks with Transformer self-attention layers, combining their strengths. We also introduce register tokens and a new SE-Scaling attention mechanism to improve performance while reducing computational costs. The resulting super-resolution network, SST (State Space Transformer), achieves state-of-the-art results on both classical and lightweight tasks.

## 1 INTRODUCTION

Single image super-resolution (SISR) aims to recover high-resolution images from their degraded low-resolution counterparts. Due to its wide range of applications, exploring efficient and effective SR algorithms has long been a prominent research topic in the field of computer vision (Jo et al., 2018; Wang et al., 2019; Anwar et al., 2020). Since the pioneering works (Dong et al., 2014; Kim et al., 2016; Zhang et al., 2018a; Ledig et al., 2017; Shi et al., 2016; Lim et al., 2017), deep neural network-based methods have become the mainstream approach for SISR. These neural networks are constructed using different building blocks, leading to various characteristics.

Convolutional Neural Networks (CNNs) use convolutional layers as their main component, processing neighboring pixels through convolutions and expanding the network's receptive field by stacking multiple convolutional layers. This practice has led to many successful SR network designs (Tai et al., 2017; Ledig et al., 2017; Lim et al., 2017; Kim et al., 2016; Zhang et al., 2021; Li et al., 2018; Wang et al., 2018; Zhang et al., 2018c; Tong et al., 2017; Zhang et al., 2018b; Yang & Qi, 2021). However, the inherent locality inductive bias in CNNs makes it difficult for these networks to effectively exploit non-local information (Shi et al., 2022). In contrast, Transformer networks (Chen et al., 2021; Liang et al., 2021; Zhang et al., 2022a;c;a;c; Chen et al., 2023a), which use self-attention mechanisms to process spatial information, have achieved success in overcoming these limitations. The self-attention mechanism of Transformers does not assume a locality inductive bias and theoretically has the ability to cover a larger receptive field, potentially leading to better SR performance. However, due to the quadratic computational complexity of the self-attention mechanism with respect to the number of tokens, in practice, we cannot equip Transformers with sufficiently large windows. Methods like SwinIR (Liang et al., 2021), which are based on shifted windows, still perform self-attention processing only locally and thus cannot effectively utilize global information.

This limitation of Transformer networks impacts not only image processing network design but also numerous other fields that rely on self-attention mechanisms and face constraints due to their quadratic complexity with respect to the number of input tokens. To alleviate this problem, the Mamba models introduce a novel State Space Model (SSM) (Gu & Dao, 2023; Mehta et al., 2022; Wang et al., 2023), offering a new method for long-sequence modeling with linear complexity, initially applied in natural language processing. Mamba models have also been successfully applied to visual tasks and image processing, including SISR, such as MambaIR (Guo et al., 2024) and DVMSR (Lei et al., 2024). By organizing pixels into long sequences in a scanning manner and

processing them using the SSM blocks, an essentially global attention mechanism is achieved. This has led to high expectations that Mamba models could solve the current problems of Transformers and convolutional networks.

However, existing works have also revealed some issues with the Mamba model; they have not demonstrated significant performance advantages. Recent works, such as Vision Mamba (Zhu et al., 2024), VMamba (Liu et al., 2024), and MambaOut (Yu & Wang, 2024), have shown through experiments that vision models based on SSM, despite having larger receptive fields and lower computational costs, perform poorly on many visual tasks that do not involve long sequences when compared to state-of-the-art convolutional and attention-based models. This suggests that the scanning method of vision ssms, which traverses along the row or column axis and flattens spatial tokens into long sequences, makes it unable to capture local contextual dependencies in 2D images as efficiently as attention or convolution. As a result, their local region representation capability within their effective receptive field is inferior to that of Transformers.

In this work, we aim to leverage the stronger representation capability of Transformer models and introduce the low-complexity global information processing ability of the Mamba model into our architecture. We find that integrating the Mamba SSM as an additional module with Transformers can combine the advantages of both methods, complementing each other. We conduct an in-depth study on mixing Mamba SSM blocks with Transformer self-attention layers and propose a simple yet general model that achieves better results than both Mamba and Transformers without complex designs or a significant increase in computational complexity and parameters. Furthermore, we investigate the reasons behind the weak local region representation capabilities of vision Mamba models and propose solutions. Our results indicate that the internal modeling of Mamba exhibits significant problems when processing visual inputs. Specifically, Mamba model generates feature maps with many artifacts; these artifacts correspond to abnormal tokens with unusually high regularization values, and these tokens tend to discard local information in favor of containing more global information. These abnormal artifacts greatly affect the quality of the feature maps. To fundamentally address this deficiency in vision Mamba networks, we propose adding updatable register tokens in vision ssms that are independent of the input tokens. Additionally, works like MambaIR have shown that introducing channel attention mechanisms can improve performance, but this method introduces a substantial additional computational burden. We propose a new attention mechanism, SE-Scaling, to replace channel attention, achieving better improvements while significantly reducing the computational cost. By integrating the above methods, we propose a super-resolution network called SST (State Space Transformer), which achieves state-of-the-art performance on both classical and lightweight tasks.

## 2 RELATED WORK

**Vision Transformer.** Transformers have recently shown great potential in various visual tasks, including image restoration tasks (Zamir et al., 2022; Liang et al., 2021; Chen et al., 2021). Among them, the most typical work should be Vision Transformer (ViT) (Dosovitskiy, 2020) , which proves that Transformers outperform convolutional neural networks in feature encoding. Image super-resolution is an important task in image restoration, and Transformer-based models also dominate. IPT (Chen et al., 2021) is a large pre-trained model based on the Transformer encoder and decoder structure, which has been applied to tasks such as super-resolution, denoising, and rain removal. Based on the Swin Transformer encoder (Liu et al., 2021), SwinIR (Liang et al., 2021) performs self-attention calculations on N×N local windows during feature extraction, achieving outstanding performance. However, existing works have not been able to solve the problem that Transformers are limited by computational complexity, which results in only utilizing limited spatial information. Existing methods, such as ELAN (Zhang et al., 2022c), simplify the architecture of SwinIR and use self-attention with different window sizes to capture correlations between distant pixels, but this also sacrifices some of the original model's representation capability in local regions. Our work retains the advantages of window self-attention in local areas while efficiently utilizing more global information for image super-resolution.

**State-Space Model.** State-space models (SSMs) (Gu et al., 2021a;b); (Smith et al., 2022) originated from classical control theory (Kalman, 1960) and have recently been introduced into deep learning as a competitive backbone for state-space transformation. In modeling long-range dependencies, the good property of linear scaling with sequence length has attracted great interest from researchers. Recently, Mamba (Gu & Dao, 2023) is a data-dependent SSM with a selection mecha-

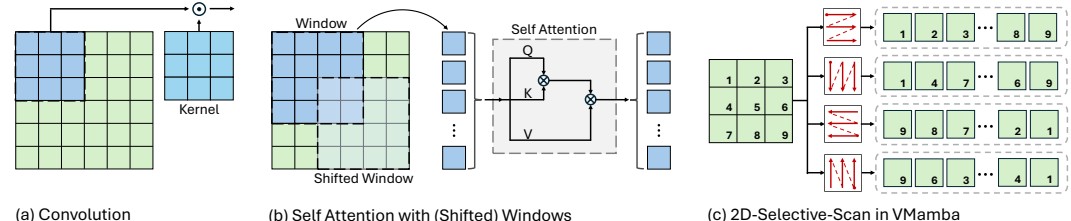

Figure 1: A diagram illustrating convolution, self-attention in Transformer networks, and the 2D-Selective-Scan mechanism in Mamba networks. It can be observed that Mamba's scanning covers more pixels but weakens the correlation between neighboring pixels.

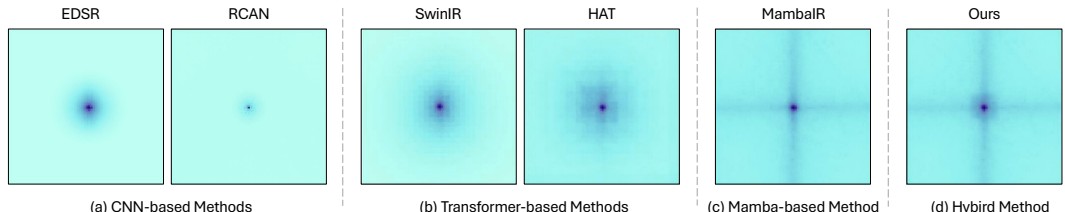

Figure 2: Visualization of the effective receptive fields of different networks. It can be seen that both convolutional and window-based Transformers can only cover a limited area, while Mamba's coverage extends across the entire image. The visualization also shows that Mamba's scanning mechanism results in a higher focus on pixels in the horizontal and vertical directions.

nism and efficient hardware design. It outperforms Transformers in natural language processing and has the property of linear scaling with input length. In addition, there are some pioneering works that apply Mamba to vision tasks such as image classification (Zhu et al., 2024), video understanding (Wang et al., 2023), and image restoration (Guo et al., 2024). However, some recent works such as Mambaout (Yu & Wang, 2024) have shown that Mamba is not suitable as a backbone for non-long sequence vision tasks, which naturally includes image super-resolution tasks, where Mamba performs poorly. In our work, Transformer and Mamba are effectively integrated, and the disadvantage of the Mamba model of losing local information when processing two-dimensional images is also compensated.

## 3 METHOD

### 3.1 MOTIVATION

In recent years, methods based on CNNs and Transformers have become mainstream in SISR tasks, especially those utilizing the Swin Transformer. However, due to the substantial computational overhead caused by the quadratic complexity of self-attention, all methods based on the Swin Transformer cannot freely expand their receptive fields and are limited to using spatial information within restricted window regions. In contrast, the Mamba model (Gu & Dao, 2023) is not constrained by quadratic complexity and can effectively utilize global information. Mamba arranges pixels in a scanning manner to form sequences and then processes them using a linear-complexity State Space Model (SSM). A comparison among Mamba, CNN, and Transformer is demonstrated in Figure 1. The visualization of the effective receptive field in Figure 2 shows that methods based on Mamba, with SSM at their core, have a wider receptive field than CNN-based and Transformer-based methods.

Despite this advantage, current SR networks based on Mamba (Guo et al., 2024) have not outperformed Transformer-based methods like SwinIR (Liang et al., 2021). The scanning approach of Mamba over pixels makes it challenging for the network to efficiently model the relationships between local pixels. Although SSM (Gu & Dao, 2023) provides a larger receptive field, it does not fully exploit the rich pixel information in practical SR tasks. Many contemporary works have confirmed this observation, such as Vision Mamba (Zhu et al., 2024), VMamba (Liu et al., 2024), and MambaOut (Yu & Wang, 2024). This naturally leads to two questions:

- Given that self-attention and SSM both have inherent shortcomings, and their advantages and disadvantages complement each other, is combining the two the optimal solution?

| Methods | Params | MACs (Flops) | Set5 | Set14 | B100 | Urban100 | Manga109 |
|---|---|---|---|---|---|---|---|
| SwinIR (Liang et al., 2021) (all MSA blocks) | 930K | 64G | 32.44 | 28.77 | 27.69 | 26.47 | 30.92 |
| MambaIR (Guo et al., 2024) (all VSSM blocks) | 979K | 57G | 32.47 | 28.80 | 27.71 | 26.55 | 31.12 |
| Combine in series, starting with MSA | 984K | 58G | 32.51 | 28.83 | 27.73 | 26.66 | 31.20 |
| Combine in parallel | 984K | 58G | 32.49 | 28.81 | 27.71 | 26.61 | 31.16 |
| Combine in series, starting with VSSM | 984K | 58G | 32.53 | 28.86 | 27.74 | 26.68 | 31.23 |

Table 1: Performance comparison of three VSSB and MSA combination methods: parallel, sequential with VSSB followed by MSA, and sequential with MSA followed by VSSB. All combinations outperform models using only MSA or VSSB, with the sequential approach of VSSB followed by MSA yielding the best results.

| | All MSA | VSSM 1:4 MSA | VSSM 1:2 MSA | VSSM 1:1 MSA | VSSM 2:1 MSA | VSSM 4:1 MSA | All VSSM |
|---|---|---|---|---|---|---|---|
| Params | 930K | 869K | 981K | 984K | 923K | 877K | 979K |
| Set5 | 32.44 | 32.44 | 32.57 | 32.53 | 32.49 | 32.43 | 32.47 |
| Set14 | 28.77 | 28.78 | 28.89 | 28.86 | 28.81 | 28.77 | 28.80 |

Table 2: The table presents the performance outcomes for various hybrid architecture designs with different ratios of VSSB to MSA blocks.

- If so, how can we maximize the benefits of their combination?

In this work, our goal is to integrate self-attention-based Transformers with SSM-based Mamba, finding the optimal way to combine them to maximize their respective strengths. We also further modify the Mamba module to enhance its effectiveness in addressing the representation capability issues in Mamba networks.

## 3.2 INTEGRATION

**Basic Structure.** Combining SSMs with self-attention is an intuitive idea, but determining the best way to integrate them requires exploratory experiment. For the SSM component, we selected the Vision State-Space Block (VSSB) used in MambaIR as the building block for the Mamba model. For the self-attention component, we chose the (Shifted) Window Multi-head Self-Attention (MSA) building block from SwinIR. This choice avoids introducing special designs, ensuring that our conclusions are generalizable. We combined VSSB and MSA in a one-to-one ratio. The methods of combining VSSB and MSA can be divided into two types: serial and parallel, with the serial combination requiring attention to the order of execution.

In Table1, we present the performance of three different combination methods on benchmark datasets. Surprisingly, all three combinations show performance improvements over models using only MSA or VSSB. This indicates that integrating Mamba and Transformer components is a promising direction. Among these, the improvement from the parallel combination is smaller compared to the serial combinations. Notably, the sequential connection where VSSB is followed by MSA achieves the best results. This suggests that we should first model the global pixel information of the input data using the state-space approach before computing self-attention in local window regions. This finding establishes the main direction of our method: combining VSSB and MSA in series and ensuring that VSSB is executed first to maximize their respective advantages.

**Finding the Optimal Integration Ratio.** Furthermore, the different structural designs combining VSSB and MSA exhibit varying performances, which indicates that they play different roles in SR networks. Designing architectures where both components are equal in quantity and connected in pairs may prevent each from fully leveraging their respective advantages. Adjusting the quantities of the two components to an optimal ratio could further enhance the capabilities of this hybrid structure.

To explore the individual influences of SSM and Self-Attention and determine the optimal quantity ratio between them, we designed five different hybrid architectures. The specific structural designs are illustrated in Figure 3(a). The experimental results, shown in Figure 3(b) and Table 2, demonstrate that combining VSSB and MSA improves performance across different ratios, with the model achieving peak performance when the ratio between the two is 1:2. This finding aligns with our earlier inference that vision Mamba cannot serve as the backbone model for SR tasks on its own. Only by integrating it with Transformers and controlling the ratio between them can vision Mamba fully maximize its advantages. The 1:2 ratio is also related to the shift-window mechanism of MSA; due to this mechanism, MSA blocks are typically grouped in pairs to achieve optimal results.

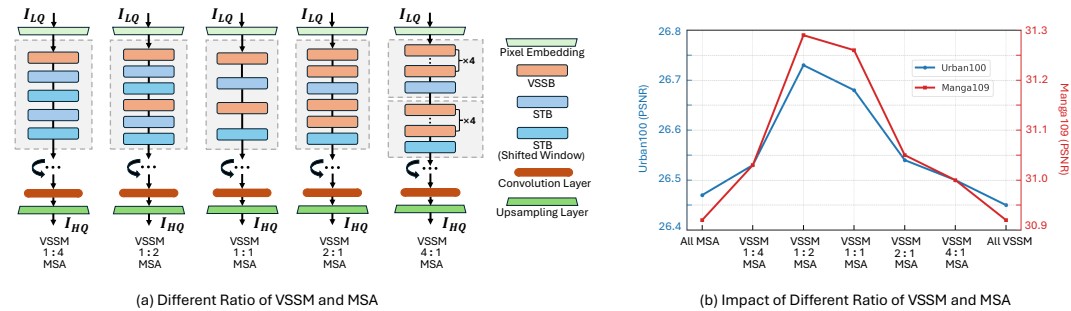

(a) Different Ratio of VSSM and MSA          (b) Impact of Different Ratio of VSSM and MSA

Figure 3: Exploration of the optimal VSSB-to-MSA ratio in hybrid architectures. (a) illustrates different structural designs and (b) shows the experimental results indicating performance improvements across different ratios, with the optimal ratio identified as 1:2.

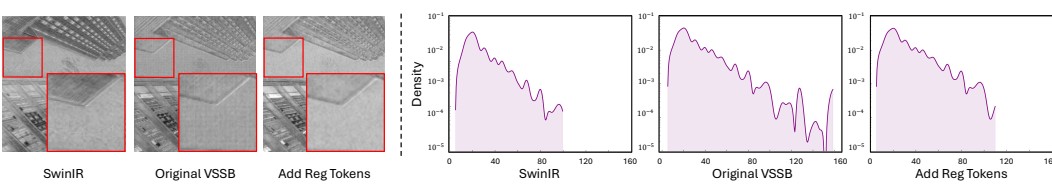

(a) Feature Artifacts           (b) L2 Norm distribution

Figure 4: Visualization of feature artifacts. (a) Feature map from the hybrid model with VSSB, showing artifacts in low-frequency regions. (b) Norm distribution of feature map tokens, revealing numerous high-norm outliers in vision Mamba.

## 3.3 FURTHER IMPROVEMENT OF VISION STATE-SPACE BLOCK

**Feature Artifacts of Vision State Space Model.** In ViT (Alexey, 2020), feature maps often contain a considerable number of outliers that correspond to low-information background regions but exhibit abnormally high attention scores. A recent study by Darcet et al. (2023) refers to these outliers as feature artifacts. Specifically, they point out that these artifact tokens always have high norm values and, during inference, tend to discard local information in favor of retaining global features, thereby compromising the quality of the feature map. These characteristics of artifact tokens align with the shortcomings we previously identified in Mamba. In the SR task, Mamba also demonstrates a loss of pixel information and weak representation of local regions in 2D images. This similarity raises the question: Could Mamba's issues be related to feature artifact tokens?

To investigate this, we conducted a quantitative analysis of the mamba building block in our hybrid architecture model and plotted the norm distribution of the feature map tokens (see Figure 4(b)). This distribution sums the norm values of feature map tokens across all channels and clearly shows numerous high-norm outliers. These results indicate that vision Mamba is also afflicted by feature artifacts. Such high-norm artifacts can adversely affect feature extraction. Additionally, by directly observing the feature map visualization in Figure 4(a), we observe that our hybrid model combined with VSSB exhibits a large number of artifacts in low-frequency areas with less information, which seriously affects the quality of the feature map. Mamba's method of scanning and flattening all spatial domain tokens inherently loses the local spatial correlations of two-dimensional images, and the presence of numerous feature artifacts that tend to abandon local information exacerbates this issue. Therefore, addressing the artifact problem is of great significance in overcoming vision Mamba's weak representation ability in two-dimensional areas.

**Introducing Register Tokens for Artifact Removal.** Building on the work by Darcet et al. (2023) that proposed a solution to remove artifacts in ViT, we address this problem by introducing register tokens into the SSM.

Our approach adds register tokens before the data is input to each SSM layer and discards them after the data is output from the SSM layer. This means the registers are updated at different SSM layers within the model. Figure 5 shows the enhanced SSM layer. This register setting strategy not only avoids additional complex tensor operations when the input data passes through VSSB and MSA but also better captures and retains important semantic information at different depths of the model.

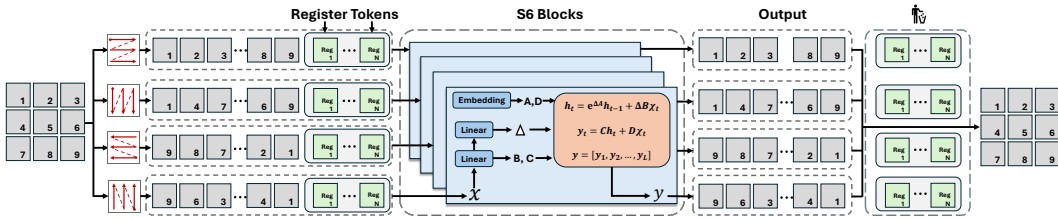

Figure 5: Illustration of vision state-space model with updateable registers. Input-independent register tokens are appended to the input data to mitigate feature artifacts. These register tokens are created before the data enters the SSM layer and are discarded upon exiting, ensuring effective artifact handling throughout the model.

| Methods | Params | Macs | Set5 | Set14 | B100 | Urban100 | Manga109 |
|---|---|---|---|---|---|---|---|
| VSSB (w/o SE-Scaling) | 1091K | 65G | 32.59 | 28.90 | 27.80 | 26.78 | 31.34 |
| VSSB (w/ Channel Attention) | 1109K | 64G | 32.58 | 28.89 | 27.80 | 26.77 | 31.34 |
| VSSB (w/ MLP) | 1063K | 64G | 32.57 | 28.88 | 27.78 | 26.73 | 31.28 |
| VSSB (w/ SE-Scaling) | 1097K | 65G | 32.63 | 28.94 | 27.81 | 26.82 | 31.41 |

Table 3: Comparison of our SE-Scaling with MLP and different attention modules. The results show that our SE-Scaling has stronger representation capabilities among models of the same size.

In our experiments, we compared the performance using different numbers of register tokens under this strategy. The results show that Figure 6 when the number of registers exceeds four, the model's performance remains almost unchanged. Since increasing the number of register tokens also increases the computational complexity, we opt to add four register tokens after the input token sequence.

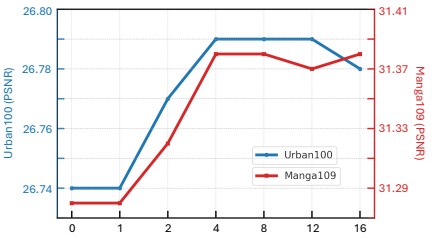

Figure 6: Performance differences due to different number of register tokens.

**SE-Scaling.** Our SE-Scaling, as shown in Figure 7(c), specifically includes two parts: a variant of channel attention (Channel Scaling) and spatial attention (Spatial Squeeze and Excitation). In Channel-Scaling, we first perform global average pooling to compress the spatial dimension of the input feature map to $1\times1$ and generate a global feature representation for each channel. This operation can be expressed as: $z_c = \frac{1}{H \times W} \sum_{i=1}^{H} \sum_{j=1}^{W} x_{b,c,i,j}$. Then use a $1\times1$ convolutional layer to map the compressed features to a single channel to obtain the excitation output $y$, and perform ReLU activation: $y_c = \text{ReLU}(W_c z_c + b_c)$, then use nearest neighbor interpolation to adjust $y$ back to the original spatial dimension, scale the original input $x$ according to the stimulus output $y$: $\hat{x}_{b,c,i,j} = x_{b,c,i,j} \cdot y_{b,c,i,j}$.

While sSE focuses on enhancing the important spatial regions in the feature map. It first applies a $1x1$ convolutional layer to the input feature map to convert the input channel into a single channel: $y_{b,1,i,j} = \sum_{c=1}^{C} W_{c,1} x_{b,c,i,j} + b$, This convolution operation captures the spatial information of all input channels. The output of the convolution is then processed through a $Sigmoid$ activation function to normalize the spatial attention map to the range of [0, 1]: $y_{b,1,i,j} = \sigma(y_{b,1,i,j})$. Finally, the original input feature map $x$ is scaled according to the spatial attention map $y$: $\hat{x}_{b,c,i,j} = x_{b,c,i,j} \cdot y_{b,1,i,j}$. Finally, Channel-Scaling and sSE are fused to take their maximum value: $\hat{x}_{b,c,i,j} = \max(\hat{x}_{b,c,i,j}^{cSE}, \hat{x}_{b,c,i,j}^{sSE})$. Our SE-Scaling can focus on spatial and channel features very efficiently, further improving the performance of the model. The results in Table 3 shows that VSSB, which replaces channel attention with SE-Scaling, achieves optimal performance under the condition of similar model size.

### 3.4 OVERALL ARCHITECTURE

The overall architecture of our State-Space Transformer SR network (SST) is depicted in Figure 7(a). The network begins with a $3\times3$ convolutional layer, which extracts initial feature maps from the input image. These features are then processed through multiple stages of our hybrid module, which consists of Vision State-Space Blocks with Registers (VSSB-R) and Swin Transformer

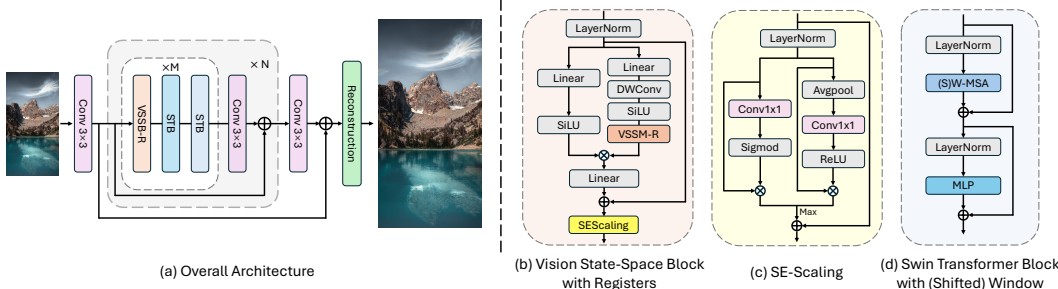

(a) Overall Architecture     (b) Vision State-Space Block with Registers     (c) SE-Scaling     (d) Swin Transformer Block with (Shifted) Window

Figure 7: (**a**) The architecture of our proposed SST for image resolution. (**b**) The inner structure of Vision State-Space Block with updateable Registers (VSSB-R). (**c**) The inner structure of SE-Scaling. (**d**) The inner structure of Swin Transformer Block with (shifted) window

Blocks (STB). The hybrid module is repeated N times to allow for deeper feature extraction and better learning of intricate patterns in the image. Within each hybrid module, the VSSB-R blocks and STBs are combined in a serial arrangement, repeated M times. This design enables the model to first leverage the global pixel information processing capabilities of VSSB-R, followed by the local spatial representation power of the STBs. The inclusion of register tokens in VSSB-R ensures the preservation of important semantic information, improving the model's handling of feature artifacts. Following the hybrid modules, another 3x3 convolutional layer is applied to refine the extracted features. A global residual connection is employed, adding the input feature map to the final feature map to aid in the recovery of fine details. Finally, a reconstruction module is used to generate the high-resolution output image. This combination of VSSB-R and STB allows the model to efficiently capture both global and local contextual information, resulting in enhanced super-resolution performance.

## 4 EXPERIMENTS

We have verified some core conclusions supporting our network structure design through some experiments. Next, we conduct experiments on both classical and lightweight image SR tasks, compare our SST with existing state-of-the-art methods.

### 4.1 EXPERIMENTAL SETTINGS

**Datasets and Evaluation.** The selection of training datasets is consistent with the comparison models. In classical image SR, we use DIV2K (Lim et al., 2017) and DF2K (DIV2K (Lim et al., 2017) + Flickr2K (Timofte et al., 2017)) to train our SST. In lightweight image SR, we use DIV2K (Lim et al., 2017) to train our SST-light. For testing, we mainly evaluate our method on five benchmark datasets, including Set5 (Bevilacqua et al., 2012), Set14 (Zeyde et al., 2012), BSD100 (Martin et al., 2001), Urban100 (Huang et al., 2015), and Manga109 (Matsui et al., 2017). The experimental results are evaluated in terms of PSNR and SSIM values, which are calculated based on the Y channel of the YCbCr space.

**Implementation Details.** In the classical image SR task, we set the Residual group number, VSSB-R number, STB number, channel number, windows size, and attention head number to 6, 2, 4, 180, 16, and 6, respectively. For the lightweight image SR task, we set the Residual group number, VSSB-R number, STB number, channel number, windows size, and attention head number to 4, 2, 4, 60, 8, and 6, respectively. The training patch size we use is $64 \times 64$. We randomly rotate images by 90°, 180°, or 270° and randomly flip images horizontally for data augmentation. We adopt the Adam Kingma (2014) optimizer with $\beta_1 = 0.9$ and $\beta_2 = 0.99$ to train the model for 500k iterations. The initial learning rate is set as $2 \times 10^{-4}$ and is reduced by half at the $\{250k, 400k, 450k, 475k\}$-th iterations.

### 4.2 CLASSICAL IMAGE SUPER-RESOLUTION

For the classical image SR task, we compare our Method with a series of state-of-the-art CNN-based, Transformer-based and Mamba-based SR methods: EDSR (Lim et al., 2017), RCAN (Zhang et al., 2018b), SAN (Dai et al., 2019), HAN (Niu et al., 2020), IPT (Chen et al., 2021), SwinIR (Liang et al., 2021), EDT (Li et al., 2021), CAT-R (Chen et al., 2022), ART-S (Zhang et al., 2022b), SR-Former (Zhou et al., 2023), DAT-S (Chen et al., 2023b), MambaIR (Guo et al., 2024).

| Method | Scale | Params | Set5 | | Set14 | | B100 | | Urban100 | | Manga109 | |
|---|---|---|---|---|---|---|---|---|---|---|---|---|
| | | | PSNR | SSIM | PSNR | SSIM | PSNR | SSIM | PSNR | SSIM | PSNR | SSIM |
| EDSR | ×2 | 42.6M | 38.11 | 0.9602 | 33.92 | 0.9195 | 32.32 | 0.9013 | 32.93 | 0.9351 | 39.10 | 0.9773 |
| RCAN | ×2 | 15.4M | 38.27 | 0.9614 | 34.12 | 0.9216 | 32.41 | 0.9027 | 33.34 | 0.9384 | 39.44 | 0.9786 |
| SAN | ×2 | 15.7M | 38.31 | 0.9620 | 34.07 | 0.9213 | 32.42 | 0.9028 | 33.10 | 0.9370 | 39.32 | 0.9792 |
| HAN | ×2 | 63.6M | 38.27 | 0.9614 | 34.16 | 0.9217 | 32.41 | 0.9027 | 33.35 | 0.9385 | 39.46 | 0.9785 |
| IPT | ×2 | 115M | 38.37 | - | 34.43 | - | 32.48 | - | 33.76 | - | - | - |
| SwinIR | ×2 | 11.8M | 38.42 | 0.9623 | 34.46 | 0.9250 | 32.53 | 0.9041 | 33.81 | 0.9433 | 39.92 | 0.9797 |
| EDT | ×2 | 11.5M | 38.45 | 0.9624 | 34.57 | 0.9258 | 32.52 | 0.9041 | 33.80 | 0.9425 | 39.93 | 0.9800 |
| CAT-R | ×2 | 16.6M | 38.48 | 0.9625 | 34.53 | 0.9251 | 32.56 | 0.9045 | 34.08 | 0.9443 | 40.09 | 0.9804 |
| ART-S | ×2 | 11.9M | 38.48 | 0.9625 | 34.50 | 0.9258 | 32.53 | 0.9043 | 34.02 | 0.9437 | 40.11 | 0.9804 |
| SRFormer | ×2 | 10.9M | 38.51 | 0.9627 | 34.44 | 0.9253 | 32.57 | 0.9046 | 34.09 | 0.9449 | 40.07 | 0.9802 |
| DAT-S | ×2 | 11.2M | 38.54 | 0.9627 | 34.60 | 0.9258 | 32.57 | 0.9047 | 34.12 | 0.9444 | 40.17 | 0.9804 |
| MambaIR | ×2 | 12.8M | 38.48 | 0.9624 | 34.55 | 0.9256 | 32.54 | 0.9045 | 33.96 | 0.9436 | 39.99 | 0.9801 |
| **SST (ours)** | ×2 | 11.4M | 38.57 | 0.9628 | 34.72 | 0.9265 | 32.58 | 0.9047 | 34.29 | 0.9452 | 40.12 | 0.9802 |
| EDSR | ×3 | 43.0M | 34.65 | 0.9280 | 30.52 | 0.8462 | 29.25 | 0.8093 | 28.80 | 0.8653 | 34.17 | 0.9476 |
| RCAN | ×3 | 15.6M | 34.74 | 0.9299 | 30.65 | 0.8482 | 29.09 | 0.8111 | 29.09 | 0.8702 | 34.44 | 0.9499 |
| SAN | ×3 | 15.9M | 34.75 | 0.9300 | 30.59 | 0.8476 | 29.33 | 0.8112 | 28.93 | 0.8671 | 34.30 | 0.9494 |
| HAN | ×3 | 64.2M | 34.75 | 0.9299 | 30.67 | 0.8483 | 29.32 | 0.8110 | 29.10 | 0.8705 | 34.48 | 0.9500 |
| IPT | ×3 | 116M | 34.81 | - | 30.85 | - | 29.38 | - | 29.49 | - | - | - |
| SwinIR | ×3 | 11.9M | 34.97 | 0.9318 | 30.93 | 0.8534 | 29.46 | 0.8145 | 29.75 | 0.8826 | 35.12 | 0.9537 |
| EDT | ×3 | 11.6M | 34.97 | 0.9316 | 30.89 | 0.8527 | 29.44 | 0.8142 | 29.72 | 0.8814 | 35.13 | 0.9534 |
| CAT-R | ×3 | 16.6M | 34.99 | 0.9320 | 31.00 | 0.8539 | 29.49 | 0.8154 | 29.91 | 0.8848 | 35.29 | 0.9542 |
| ART-S | ×3 | 11.9M | 34.98 | 0.9318 | 30.94 | 0.8530 | 29.45 | 0.8146 | 29.86 | 0.8830 | 35.22 | 0.9539 |
| SRFormer | ×3 | 10.6M | 35.02 | 0.9323 | 30.94 | 0.8540 | 29.48 | 0.8156 | 30.04 | 0.8865 | 35.26 | 0.9543 |
| DAT-S | ×3 | 11.3M | 35.12 | 0.9327 | 31.04 | 0.8543 | 29.51 | 0.8157 | 29.98 | 0.8846 | 35.41 | 0.9546 |
| MambaIR | ×3 | 12.8M | 34.97 | 0.9318 | 30.92 | 0.8534 | 29.46 | 0.8144 | 29.80 | 0.8828 | 35.20 | 0.9541 |
| **SST (ours)** | ×3 | 11.4M | 35.04 | 0.9325 | 31.04 | 0.8545 | 29.51 | 0.8159 | 30.16 | 0.8869 | 35.46 | 0.9548 |
| EDSR | ×4 | 43.0M | 32.46 | 0.8968 | 28.80 | 0.7876 | 27.71 | 0.7420 | 26.64 | 0.8033 | 31.02 | 0.9148 |
| RCAN | ×4 | 15.6M | 32.63 | 0.9002 | 28.87 | 0.7889 | 27.77 | 0.7436 | 26.82 | 0.8087 | 31.22 | 0.9173 |
| SAN | ×4 | 15.9M | 32.64 | 0.9003 | 28.92 | 0.7888 | 27.78 | 0.7436 | 26.79 | 0.8068 | 31.18 | 0.9169 |
| HAN | ×4 | 64.2M | 32.64 | 0.9002 | 28.90 | 0.7890 | 27.80 | 0.7442 | 26.85 | 0.8094 | 31.42 | 0.9177 |
| IPT | ×4 | 116M | 32.64 | - | 29.01 | - | 27.82 | - | 27.26 | - | - | - |
| SwinIR | ×3 | 11.9M | 32.92 | 0.9044 | 29.09 | 0.7950 | 27.92 | 0.7489 | 27.45 | 0.8254 | 32.03 | 0.9260 |
| EDT | ×4 | 11.6M | 32.82 | 0.9031 | 29.09 | 0.7939 | 27.91 | 0.7483 | 27.46 | 0.8246 | 32.05 | 0.9254 |
| CAT-R | ×4 | 16.6M | 32.89 | 0.9044 | 29.13 | 0.7955 | 27.95 | 0.7500 | 27.62 | 0.8292 | 32.16 | 0.9269 |
| ART-S | ×4 | 11.9M | 32.86 | 0.9029 | 29.09 | 0.7942 | 27.91 | 0.7489 | 27.54 | 0.8261 | 32.13 | 0.9263 |
| SRFormer | ×4 | 10.3M | 32.93 | 0.9041 | 29.08 | 0.7953 | 27.94 | 0.7502 | 27.68 | 0.8311 | 32.21 | 0.9271 |
| DAT-S | ×4 | 11.3M | 33.00 | 0.9047 | 29.20 | 0.7962 | 27.97 | 0.7502 | 27.68 | 0.8300 | 32.33 | 0.9278 |
| MambaIR | ×4 | 12.9M | 32.93 | 0.9044 | 29.10 | 0.7952 | 27.92 | 0.7490 | 27.50 | 0.8261 | 32.08 | 0.9265 |
| **SST (ours)** | ×4 | 11.5M | 33.00 | 0.9050 | 29.20 | 0.7967 | 27.98 | 0.7505 | 27.84 | 0.8325 | 32.37 | 0.9279 |

Table 4: PSNR(dB)/SSIM comparison for **classical** image super-resolution task on five benchmark datasets. We color best and second best results in red and blue.

**Quantitative comparison.** The quantitative comparison of the methods for classical image SR is shown in Table 4. We can see that our method achieves the best performance on all five datasets. Especially on the Urban100 dataset, our model performs even better, with a minimum of $0.34$dB and a maximum of $0.46$dB improvement on three tasks compared to our baseline: SwinIR (Liang et al., 2021). This shows that our method can capture more global information than previous Transformer-based models, which is very effective for images in Urban100 with a large number of repeated texture structures.

**Qualitative comparison.** We show qualitative comparisons with other methods in Fig. 8. From the first example in Fig. 8, we can clearly observe that only our model can restore clear and detailed edges, while other models not only cannot restore clear edges, but also distort the original shape of the image. For the second example, our model is also the only one that can fully restore the cross pattern in the image. Qualitative comparison shows that our SST can restore better high-resolution images from low-resolution images.

**Model Size Comparisons.** In Table 6, we further compare our method with several image SR methods in terms of computational complexity (e.g., FLOPs), number of parameters, and performance at ×4 scale. We set the output size to 3×512×512 to calculate FLOPs, and use PSNR tested on Urban100 to evaluate the performance. Compared with our baseline: SwinIR (Liang et al., 2021), our method achieves up to 0.39 dB improvement under the condition of comparable number of parameters and computation, and at least 0.16 dB improvement compared with the most advanced Transformer-based methods such as ART, CAT-R, SRFormer, DAT-S. Such results fully demonstrate that our method of integrating SSM with Transformer is extremely effective. The combination with SwinIR alone can achieve state-of-the-art performance, and our method still has great potential.

## 4.3 LIGHTWEIGHT IMAGE SUPER-RESOLUTION

Our method not only excels in classical SR task but also demonstrates even stronger performance in lightweight task. Across all benchmarks, our method outperforms many state-of-the-art methods by a significant margin while using much less computational power. We also compare our Method with a series of state-of-the-art CNN-based, Transformer-based and Mamba-based SR methods: CARN (Ahn et al., 2018), IMDN (Hui et al., 2019), LAPAR-A (Li et al., 2020), LatticeNet (Luo

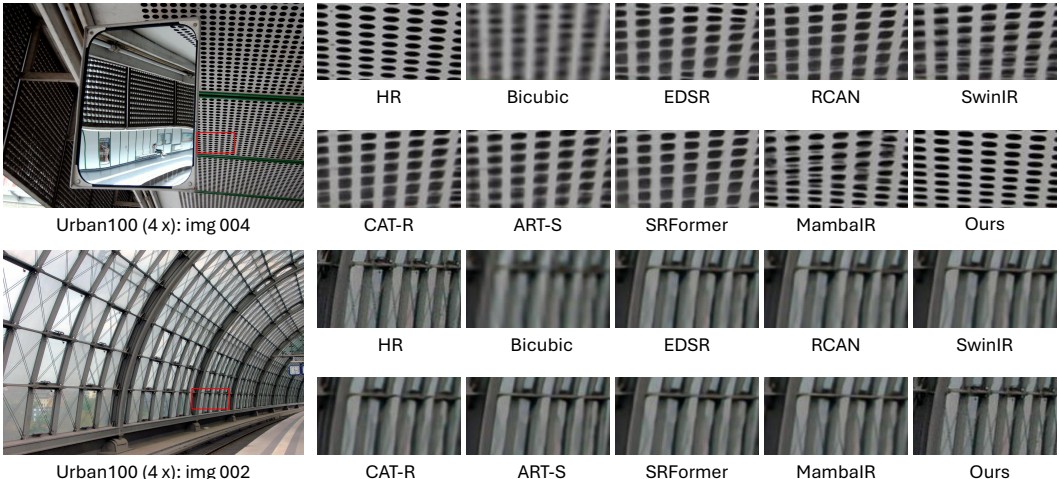

Figure 8: Qualitative comparison with recent state-of-the-art **classical image SR** methods on the ×4 SR task.

| Method | Scale | Params | Macs | Set5 | | Set14 | | B100 | | Urban100 | | Manga109 | |
|---|---|---|---|---|---|---|---|---|---|---|---|---|---|
| | | | | PSNR | SSIM | PSNR | SSIM | PSNR | SSIM | PSNR | SSIM | PSNR | SSIM |
| CARN | ×2 | 1592K | 222.8G | 37.76 | 0.9590 | 33.52 | 0.9166 | 32.09 | 0.8978 | 31.92 | 0.9256 | 38.36 | 0.9765 |
| IMDN | ×2 | 694K | 158.8G | 38.00 | 0.9605 | 33.63 | 0.9177 | 32.19 | 0.8996 | 32.17 | 0.9283 | 38.88 | 0.9774 |
| LAPAR-A | ×2 | 548K | 171G | 38.01 | 0.9605 | 33.62 | 0.9183 | 32.19 | 0.8999 | 32.10 | 0.9283 | 38.67 | 0.9772 |
| LatticeNet | ×2 | 756K | 169.5G | 38.15 | 0.9610 | 33.78 | 0.9193 | 32.25 | 0.9005 | 32.43 | 0.9302 | N/A | N/A |
| SwinIR-light | ×2 | 910K | 244G | 38.14 | 0.9611 | 33.86 | 0.9206 | 32.31 | 0.9012 | 32.76 | 0.9340 | 39.12 | 0.9783 |
| ELAN | ×2 | 621K | 203G | 38.17 | 0.9611 | 33.94 | 0.9207 | 32.30 | 0.9012 | 32.76 | 0.9340 | 39.11 | 0.9782 |
| SwinIR-NG | ×2 | 1181K | 274.1G | 38.17 | 0.9612 | 33.94 | 0.9205 | 32.31 | 0.9013 | 32.78 | 0.9340 | 39.20 | 0.9781 |
| SRFormer-light | ×2 | 853K | 236G | 38.23 | 0.9613 | 33.94 | 0.9209 | 32.36 | 0.9019 | 32.91 | 0.9353 | 39.28 | 0.9785 |
| MambaIR | ×2 | 1357K | 302G | 38.16 | 0.9610 | 34.00 | 0.9212 | 32.34 | 0.9017 | 32.92 | 0.9356 | 39.31 | 0.9779 |
| **SST-light (ours)** | ×2 | 967K | 229G | 38.23 | 0.9619 | 34.08 | 0.9233 | 32.37 | 0.9021 | 33.14 | 0.9368 | 39.39 | 0.9786 |
| CARN | ×3 | 1592K | 118.8G | 34.29 | 0.9255 | 30.29 | 0.8407 | 29.06 | 0.8034 | 28.06 | 0.8493 | 33.50 | 0.9440 |
| IMDN | ×3 | 703K | 71.5G | 34.36 | 0.9270 | 30.32 | 0.8417 | 29.09 | 0.8046 | 28.17 | 0.8519 | 33.61 | 0.9445 |
| LAPAR-A | ×3 | 594K | 114G | 34.36 | 0.9267 | 30.34 | 0.8421 | 29.11 | 0.8054 | 28.15 | 0.8523 | 33.51 | 0.9441 |
| LatticeNet | ×3 | 765K | 76.3G | 34.53 | 0.9281 | 30.39 | 0.8424 | 29.15 | 0.8059 | 28.33 | 0.8538 | N/A | N/A |
| SwinIR-light | ×3 | 918K | 111G | 34.62 | 0.9289 | 30.54 | 0.8463 | 29.20 | 0.8082 | 28.66 | 0.8624 | 33.98 | 0.9478 |
| ELAN | ×3 | 629K | 90.1G | 34.61 | 0.9288 | 30.55 | 0.8463 | 29.21 | 0.8081 | 28.69 | 0.8624 | 34.00 | 0.9478 |
| SwinIR-NG | ×3 | 1190K | 114.1G | 34.64 | 0.9293 | 30.58 | 0.8471 | 29.24 | 0.8090 | 28.75 | 0.8639 | 34.22 | 0.9488 |
| SRFormer-light | ×3 | 861K | 105G | 34.67 | 0.9296 | 30.57 | 0.8469 | 29.26 | 0.8099 | 28.81 | 0.8655 | 34.19 | 0.9489 |
| MambaIR | ×3 | 1365K | 129G | 34.72 | 0.9296 | 30.63 | 0.8475 | 29.29 | 0.8099 | 29.00 | 0.8689 | 34.39 | 0.9495 |
| **SST-light (ours)** | ×3 | 976K | 101G | 34.70 | 0.9298 | 30.67 | 0.8483 | 29.30 | 0.8103 | 29.01 | 0.8682 | 34.47 | 0.9503 |
| CARN | ×4 | 1592K | 90.9G | 32.13 | 0.8937 | 28.60 | 0.7806 | 27.58 | 0.7349 | 26.07 | 0.7837 | 30.47 | 0.9084 |
| IMDN | ×4 | 715K | 40.9G | 32.21 | 0.8948 | 28.58 | 0.7811 | 27.56 | 0.7353 | 26.04 | 0.7838 | 30.45 | 0.9075 |
| LAPAR-A | ×4 | 659K | 94G | 32.15 | 0.8944 | 28.61 | 0.7818 | 27.61 | 0.7366 | 26.14 | 0.7871 | 30.42 | 0.9074 |
| LatticeNet | ×4 | 777K | 43.6G | 32.30 | 0.8962 | 28.68 | 0.7830 | 27.62 | 0.7367 | 26.25 | 0.7873 | N/A | N/A |
| SwinIR-light | ×4 | 930K | 63.6G | 32.44 | 0.8976 | 28.77 | 0.7858 | 27.69 | 0.7406 | 26.47 | 0.7980 | 30.92 | 0.9151 |
| ELAN | ×4 | 640K | 54.1G | 32.43 | 0.8975 | 28.78 | 0.7858 | 27.69 | 0.7406 | 26.54 | 0.7982 | 30.92 | 0.9150 |
| SwinIR-NG | ×4 | 1201K | 63.0G | 32.44 | 0.8980 | 28.83 | 0.7870 | 27.73 | 0.7418 | 26.61 | 0.8010 | 31.09 | 0.9161 |
| SRFormer-light | ×4 | 873K | 62.8G | 32.51 | 0.8988 | 28.82 | 0.7872 | 27.73 | 0.7422 | 26.67 | 0.8032 | 31.17 | 0.9165 |
| MambaIR | ×4 | 1374K | 85.8G | 32.51 | 0.8993 | 28.85 | 0.7876 | 27.75 | 0.7423 | 26.75 | 0.8051 | 31.26 | 0.9175 |
| **SST-light (ours)** | ×4 | 986K | 60.8G | 32.62 | 0.9002 | 28.93 | 0.7894 | 27.79 | 0.7438 | 26.80 | 0.8068 | 31.41 | 0.9184 |

Table 5: PSNR(dB)/SSIM comparison for **lightweight** image super-resolution task on five benchmark datasets. We color best and second best results in red and blue.

et al., 2020), SwinIR-light (Liang et al., 2021), ELAN (Zhang et al., 2022c), SwinIR-NG (Choi et al., 2023), SRFormer-light (Zhou et al., 2023), MambaIR (Guo et al., 2024).

**Quantitative comparison.** Table 5 shows the quantitative comparison of lightweight image SR models. We report the MAC by upscaling low-resolution images to 1280 × 720 resolution at all scales. We can see that our SST-light achieves the best performance on all scale factors with fewer MACs on all five benchmark datasets. Compared with SwinIR and recent state-of-the-art lightweight models such as SRFormer and MambaIR, our SST-light uses less computation and achieves a huge performance lead. On both x3 and x4 tasks, our method achieves an amazing improvement of up to 0.49dB on Manga109 compared to SwinIR. This shows that our method is extremely versatile and is not only applicable to classic SR tasks that require a lot of computational resources, but also has outstanding performance on lightweight SR tasks.

**Qualitative comparison.** In Fig. 9, we qualitatively compare our SST-light with the state-of-the-art lightweight image SR models. Notably, SST-light is the only model that can clearly recover the line details in the example, and also does not have the large-area artifacts in the examples of the

| Methods | EDSR | RCAN | SwinIR | CAT-R | ART-S | SRFormer | DAT-S | MambaIR | SST (ours) |
|---|---|---|---|---|---|---|---|---|---|
| PSNR (dB) | 26.64 | 26.82 | 27.45 | 27.62 | 27.54 | 27.68 | 27.68 | 27.50 | 27.84 |
| Flops (G) | 823.3 | 261.0 | 215.3 | 292.7 | 251.2 | 206.1 | 203.3 | 197.8 | 224.6 |
| Parameters (M) | 43.1 | 15.6 | 11.9 | 16.6 | 11.9 | 10.4 | 11.2 | 12.9 | 11.5 |

Table 6: Table 6 shows a comparison of the performance, computational complexity, and number of parameters for the image SRs. FLOPs are measured with the output size set to $3 \times 512 \times 512$, and PSNR values are tested on Urban100 ($\times 4$).

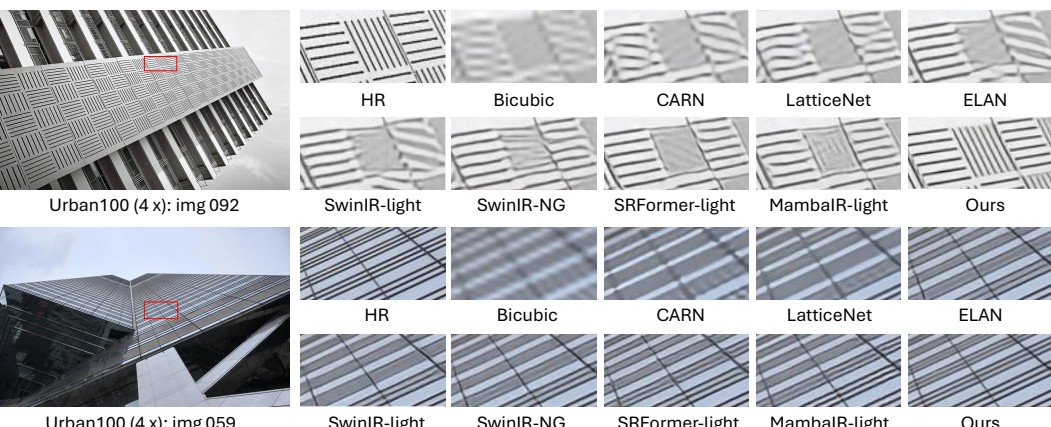

Figure 9: Qualitative comparison with recent state-of-the-art **lightweight image SR** methods for the $\times 4$ SR task.

remaining models. This strongly proves that the lightweight version of SST also performs very well in recovering edges and textures compared to other methods.

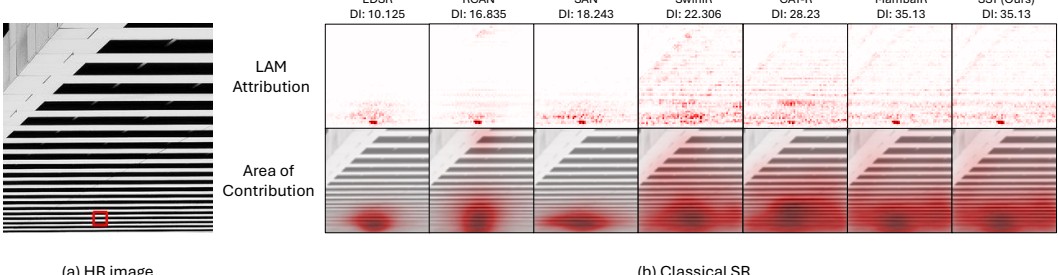

Figure 10: LAM results of SST. We can see that SST can perform SR reconstruction based on a particularly wide range of pixels compared to the other methods.

**LAM Comparison.** In Fig. 10, We can observe the range of pixels used for SR reconstruction, and we use LAM (Gu & Dong, 2021) to compare our model with many state-of-the-art methods. Based on the global receptive field brought by Mamba, the pixel range of the SR image inferred by SST is much wider than that of various Transformer-based models. The experimental results are very consistent with our motivation and demonstrate the superiority of our method from the perspective of interpretability.

## 5 CONCLUSION

In this paper, we conducted an in-depth study on mixing Mamba SSM blocks with Transformer self-attention layers. After that, we discovered the feature map artifact problem of vision Mamba and proposed to add an updateable register to solve it. Combined with our new lightweight and efficient attention mechanism SE-Scaling, we designed a very simple and highly versatile single image super-resolution model. Due to its global effective receptive field and maximum preservation of spatial correlation in two-dimensional local areas, our hybrid model SST achieves state-of-the-art performance on classic and lightweight SR tasks. We hope that our method can become a paradigm for hybrid models and a useful tool for future research on super-resolution model design.

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

## A MORE ARTIFACTS OF SST WITHOUT REGISTERS

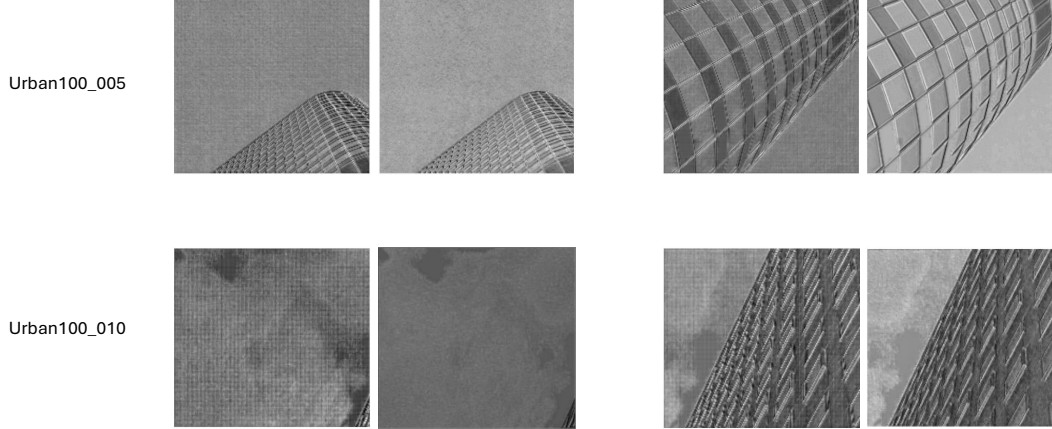

Figure 11: Each set of examples contains two pictures. The first one is the feature map of the SST model without registers, and the second one is the feature map of the SST model with registers added.

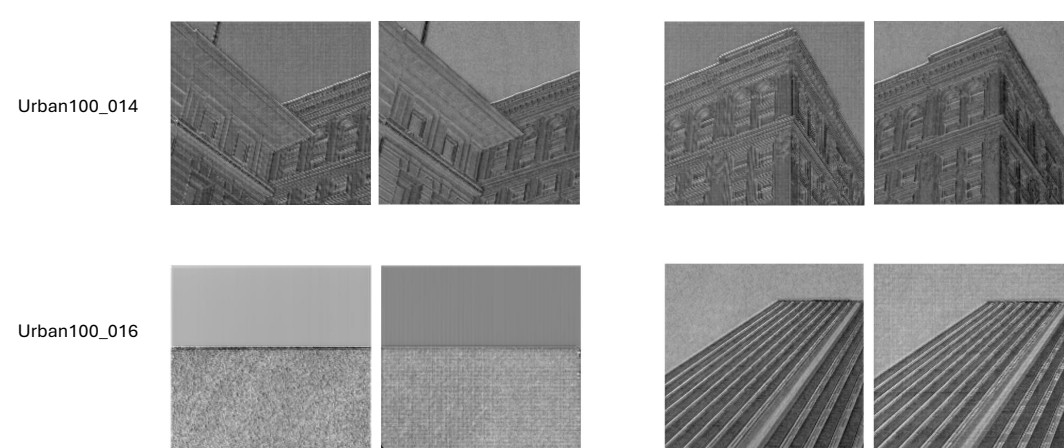

Figure 12: Each set of examples contains two pictures. The first one is the feature map of the SST model without registers, and the second one is the feature map of the SST model with registers added.

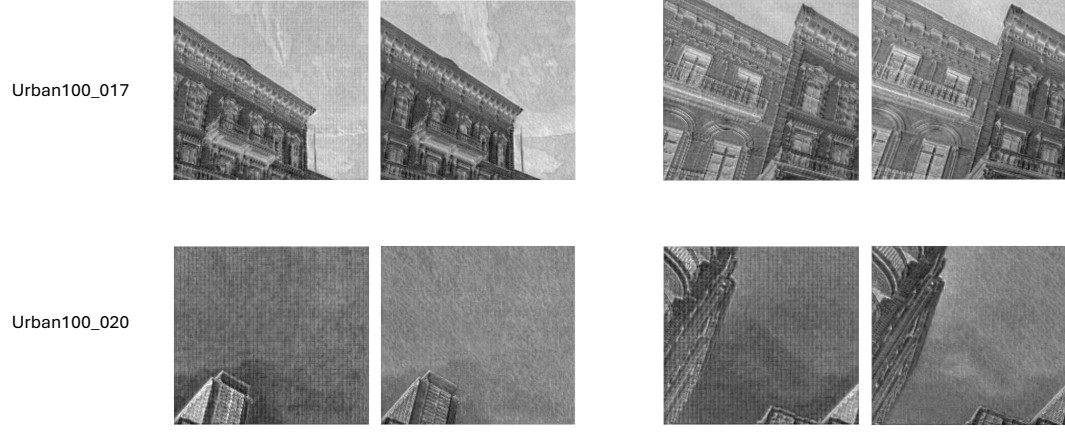

Figure 13: Each set of examples contains two pictures. The first one is the feature map of the SST model without registers, and the second one is the feature map of the SST model with registers added.

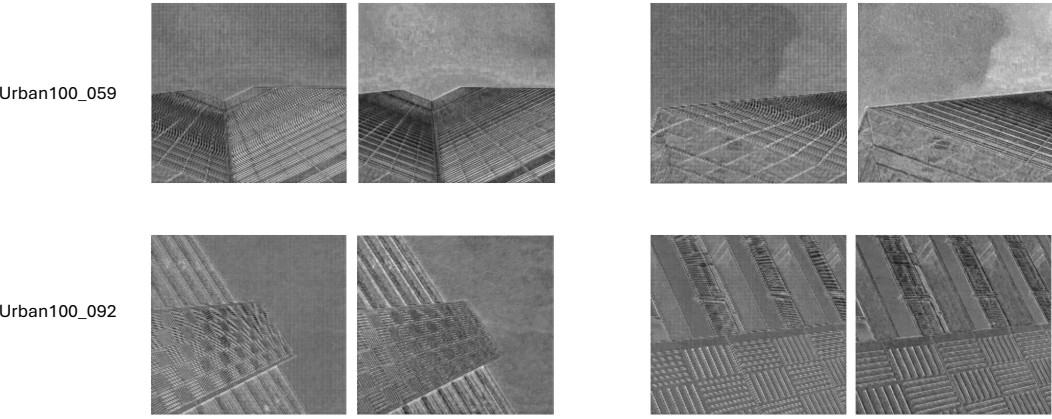

Figure 14: Each set of examples contains two pictures. The first one is the feature map of the SST model without registers, and the second one is the feature map of the SST model with registers added.

