# OpenReview forum: "Integrating State Space Model and Transformer for Global-Local Processing in Super-Resolution Networks"
_ICLR.cc/2025/Conference — ICLR 2025 Conference Withdrawn Submission_

### Official Review · Reviewer_4SuC · 2024-10-30

**Soundness:** 3
**Presentation:** 3
**Contribution:** 2
**Rating:** 3
**Confidence:** 5

**Summary:**

The paper presents a super-resolution network called SST (State Space Transformer), which integrates Mamba State Space Models (SSM) with Transformer self-attention layers. The authors aim to leverage the strengths of both architectures to enhance single image super-resolution (SISR) tasks. The Mamba model is noted for its ability to process global information efficiently due to its linear complexity, while Transformer models, particularly the Swin Transformer, excel in local region representation but suffer from quadratic complexity, limiting their receptive fields. The proposed SST model addresses the shortcomings of both approaches by combining their advantages. The authors introduce an updateable register to mitigate feature map artifacts commonly found in Mamba models and propose a new attention mechanism called SE-Scaling to reduce computational costs while improving performance.

**Strengths:**

1.	By introducing the SE-Scaling mechanism, the model reduces the computational burden typically associated with channel attention mechanisms, making it suitable for lightweight applications.
2.	The paper reports that SST achieves state-of-the-art results on both classical and lightweight super-resolution tasks, demonstrating its effectiveness through extensive experiments on various benchmark datasets.

**Weaknesses:**

1.	The combination of Mamba SSM and Transformer architectures allows the model to capture both global and local contextual information effectively, however, they are both from current knowledge. Besides, the idea of combining Mamba and Transformer has been already proposed.
2.	Compared to SOTA methods, the improvement is not significant, such as ERF in figure 2 against MambaIR and quantitative results against DAT.
3.	Missing in-depth motivation. This article seems to be just a simple attempt at an IR mission by combining Mamba and current Transformer structures.
4.	The model complexity is still large, what is the merit by using Mamba?
5.	More recent mamba-based image SR works should be referenced and compared.

**Questions:**

See the weakness.

---

### Official Review · Reviewer_NxMS · 2024-11-03

**Soundness:** 3
**Presentation:** 3
**Contribution:** 3
**Rating:** 5
**Confidence:** 4

**Summary:**

The paper proposes a single-image super-resolution network called SST, which combines Mamba and Window Attention mechanisms. The authors observed that some existing Mamba models do not effectively capture local dependencies in 2D images. Therefore, they leverage window attention to address these limitations. Additionally, the authors also observed that Mamba models tend to produce artifacts. To mitigate this issue, they introduced registration tokens before the SSM scan in SST. The authors conducted extensive experiments to explore different combinations of window attention and Mamba, and compared their method with current mainstream super-resolution networks to validate its effectiveness.

**Strengths:**

1. The paper is very easy to follow.
2. Extensive experiments were conducted to explore combinations of Mamba and Window Attention.
3. Experiments were also conducted to investigate the impact of the number of registration tokens on model performance.

**Weaknesses:**

1. Lack of comparison with some closely related approaches. With a larger number of parameters, SST-light shows lower performance metrics on almost all benchmarks compared to ATD-light[1].
2. Super resolution is a very local computation, (at the range of a pixel). It is not demonstrated what is the advantage of exploring global interaction for such a problem.

[1] Transcending the Limit of Local Window: Advanced Super-Resolution Transformer with Adaptive Token Dictionary, CVPR2024

**Questions:**

1. The impact of different proportions of VSSM and MSA on model performance was only verified on Urban100 and Manga109. It remains uncertain whether similar results would be observed on the other three datasets.
2. Similar to the previous question, it is uncertain whether the number of registration tokens also produces similar results on the other three datasets.
3.  In paper [1], it is noted that Vision Transformer has artifact issues. This raises the question of whether the window attention in this paper exhibits similar phenomena and whether it also utilizes registration tokens.

[1] Vision Transformers Need Registers, ICLR2024

---

### Official Review · Reviewer_U5Ra · 2024-11-03

**Soundness:** 2
**Presentation:** 2
**Contribution:** 2
**Rating:** 3
**Confidence:** 3

**Summary:**

This work introduces SST, a new model that integrates Mamba and Transformer to extract global and local information, respectively. This work is well-written, and the method is clear and easy to understand.

**Strengths:**

1. This work effectively integrates Mamba and Transformer, and the visualization results show that the hybrid structure can activate a wider range of pixels.
2. This work made some improvements to Mamba to alleviate the feature artifact problem.

**Weaknesses:**

1. The authors claim the proposed SE-Scaling can significantly reduce the computational cost, but in Table 3, the Macs of using SE-Scaling are higher than Channel-Attention.
2. Although this work integrates Mamba and Transformer, the proposed network SST simply uses Mamba and Transformer alternately and lacks deeper exploration.
3. The performance of SST shows only a slight improvement compared to existing state-of-the-art models, such as SRFormer.  The author should add comparisons with HAT, OmniSR, etc.

**Questions:**

Why do the authors not compare their method with HAT and discuss the advantages and disadvantages between them?

---

### Official Review · Reviewer_ttNo · 2024-11-04

**Soundness:** 3
**Presentation:** 4
**Contribution:** 2
**Rating:** 5
**Confidence:** 5

**Summary:**

This paper presents a hybrid network based on Mamba and Transformer for image super-resolution. Register tokens and SE-Scaling attention mechanisms are introduced to improve performance and reduce computation. The experimental results demonstrated the effectiveness of the method.

**Strengths:**

1. This paper is well written and organized.

2. The combination of Mamba and Transformer is promising for improving the performance of image super-resolution tasks.

3. The ablation and main experiments are extensive and comprehensive.

**Weaknesses:**

1. The novelty of this paper is limited, and the main contribution seems to be just combining Mamba and Transformer. SE-Scaling did not show significant improvement over previous work.

2. Since Mamba models perform poorly in capturing local information, why not integrate Mamba and CNNs which are good at local modeling.

3. In addition to the parameters and FLOPs, it is necessary to compare the inference latency of the different methods on the device.

4. From Figure 10, there is no significant difference between the proposed SST and MambaIR on the LAM attribution map. Does this indicate that the Transformer provides limited benefit?

**Questions:**

Please see weakness.

---

### Note · Authors · 2024-11-12

I have read and agree with the venue's withdrawal policy on behalf of myself and my co-authors.